# Light, Sleep and Performance in Diurnal Birds

**Anne E. Aulsebrook** [1,2,*], **Robin D. Johnsson** [2]  and **John A. Lesku** [2]

1    School of BioSciences, The University of Melbourne, Melbourne, VIC 3010, Australia
2    School of Life Sciences, La Trobe University, Melbourne, VIC 3086, Australia;
     r.johnsson@latrobe.edu.au (R.D.J.); j.lesku@latrobe.edu.au (J.A.L.)
*    Correspondence: aulsebrooka@gmail.com

**Abstract:** Sleep has a multitude of benefits and is generally considered necessary for optimal performance. Disruption of sleep by extended photoperiods, moonlight and artificial light could therefore impair performance in humans and non-human animals alike. Here, we review the evidence for effects of light on sleep and subsequent performance in birds. There is accumulating evidence that exposure to natural and artificial sources of light regulates and suppresses sleep in diurnal birds. Sleep also benefits avian cognitive performance, including during early development. Nevertheless, multiple studies suggest that light can prolong wakefulness in birds without impairing performance. Although there is still limited research on this topic, these results raise intriguing questions about the adaptive value of sleep. Further research into the links between light, sleep and performance, including the underlying mechanisms and consequences for fitness, could shed new light on sleep evolution and urban ecology.

**Keywords:** artificial light at night; avian; circadian; cognition; EEG; learning; memory; performance; photoperiod; sleep

## 1. Introduction

Sleep serves essential functions. Accordingly, sleep appears to be evolutionarily conserved across distantly-related animals [1,2], where it often occupies a large proportion of an animal's life [3]. The persistence of sleep in life-threatening situations [4], such as under the risk of predation [5] and in the face of competing (waking) demands [6–8] further indicates that sleep serves important biological functions. Such functions can relate to energy homeostasis [9,10], immune system maintenance [11,12], early brain development, memory processing and learning [13–17]. Thus, interference with normal sleep patterns often impairs waking performance [18–20].

One concerning trend that could pose a threat to sleep is the increasing proliferation of artificial light at night [21,22]. Exposure to light at night from bedside lamps and electronic devices can keep humans awake and alter sleep structure [23–25]. The disruption of sleep can subsequently impair alertness and task performance during the day [26]. Furthermore, artificial lighting is not restricted to homes and offices. Artificial light at night encroaches into gardens, parks and even protected reserves [27], masking natural cycles of light and darkness [28]. A question that arises then is whether artificial light at night impairs sleep and performance in other species as well.

The influence of light on avian sleep and performance is of particular interest from both an evolutionary and ecological perspective. Avian sleep shows some key similarities to sleep in mammals, as well as important distinctions [29]. Studying sleep in birds can therefore offer insights into the evolutionary origins and functions of sleep [30]. Birds also occupy a wide range of environments, including places that are dark at night, locations exposed to constant natural light (such as the high Arctic) and brightly lit cities. In addition, some birds exhibit similar cognitive abilities to mammals despite having different brain structures [31]. At least some of these cognitive abilities are dependent on prior sleep [32].

Here, we review the evidence for effects of light on sleep and sleep-dependent performance in birds. As little is known about these effects on nocturnal birds, we necessarily focus on birds that sleep predominantly at night. First, we introduce avian sleep and how it compares to mammalian sleep. Next, we outline the evidence for effects of light (from both natural and artificial sources) on avian sleep, focusing primarily on light exposures that birds might experience in the wild. We then discuss the importance of sleep for avian cognition and performance. Finally, we describe studies that integrate light, sleep and performance in birds (directly or indirectly) while highlighting key gaps and opportunities for future research.

## 2. Characterizing Avian Sleep

Birds, like mammals, have two main types of sleep: rapid eye movement (REM) sleep and non-REM sleep [33]. These two states can be distinguished from each other and from wakefulness using brain activity (based on the electroencephalogram, or EEG), muscle tone (electromyogram, EMG) and behavior (accelerometry and/or video recordings) (Figure 1). Compared with mammals, birds have shorter episodes or "bouts" of REM sleep, rarely exceeding 16 s [34]. Historically, birds have been thought to have less REM sleep than mammals [35]; however, more recent studies have found that some birds devote more than 15% of total sleep to REM sleep [36]. Unlike mammals, birds do not show thalamocortical spindles or hippocampal sharp-wave ripples during non-REM sleep, nor do they show a hippocampal theta rhythm during REM sleep [37,38]. However, like mammals, birds that consolidate their wakefulness show decreasing slow-wave activity (typically <4.5 Hz power density) during non-REM sleep [39]. In both birds and mammals, sleep is homeostatically regulated. Sleep that is lost can be recovered by sleeping more; lost non-REM sleep can also be recovered by sleeping more intensely, as indicated by increased slow-wave activity in the EEG [39,40].

Drawing conclusions about sleep from behavior alone, i.e., without the EEG, can be misleading [41]. Sleeping birds can exhibit behavior that might seem incompatible with sleep. For example, during non-REM sleep, ostriches (*Struthio camelus*) sit upright with both eyes open [42]; great frigatebirds (*Fregata minor*) can even engage in non-REM and REM sleep while flying [7]. Behavioral studies of sleep can also miss important insights into the composition and intensity of sleep [41]. Nevertheless, recording the EEG can be difficult or impossible in some study contexts. Some birds, such as great tits (*Parus major*), are too small to bear the weight of even the smallest EEG data loggers [43]. Others, such as water birds, live in environments where successfully deploying and retrieving data loggers is challenging [44]. Although advances in technology are gradually overcoming these issues, existing studies have often relied exclusively on behavior to characterize avian sleep. Consequently, although we focus primarily on EEG-based sleep studies in this review, we often refer to behavioral studies where literature is scarce.

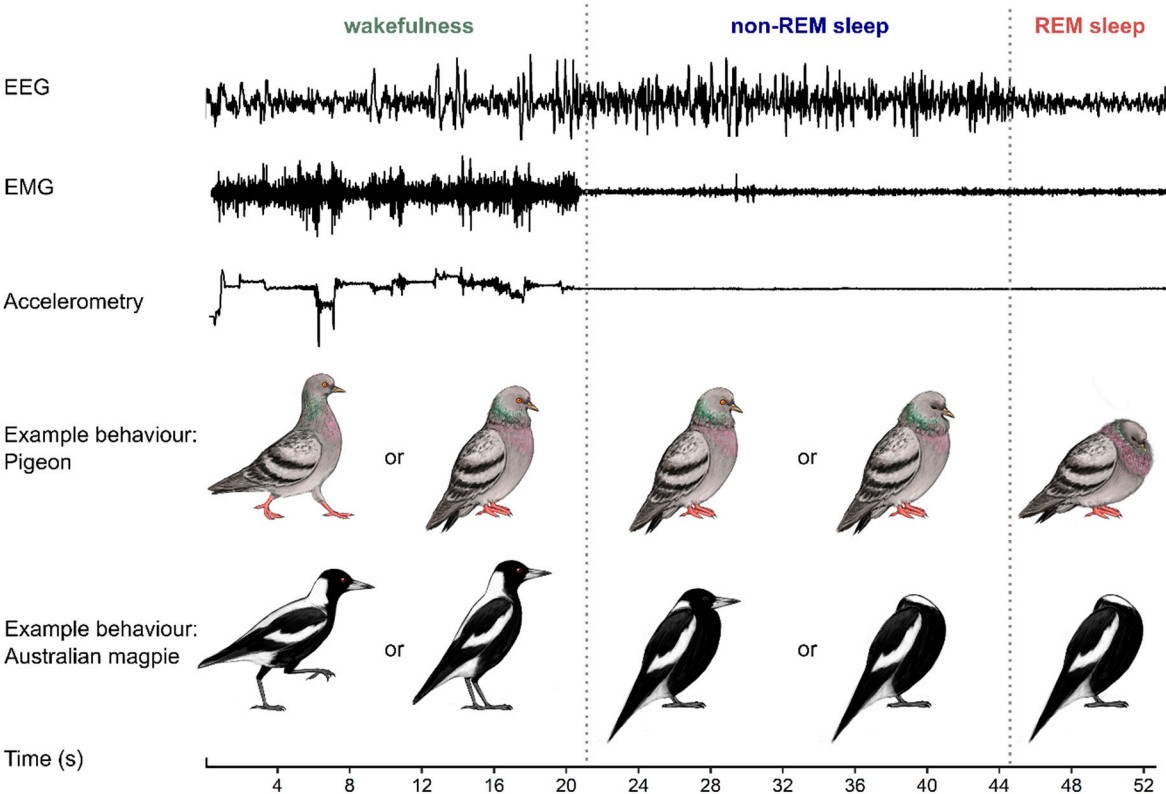

**Figure 1.** Birds can quickly transition between states of wakefulness, non-rapid eye movement (non-REM) sleep and REM sleep. When birds are awake, the electroencephalogram (EEG) is activated, muscle tone is typically high and variable (shown by the electromyogram or EMG), and the bird is often moving (shown by recordings of accelerometry or video). Non-REM sleep is characterized by slow (<4 Hz) large waves in the EEG, typically accompanied by relaxed skeletal musculature and quiescent behavior; many birds (including pigeons) can also have one or both eyes open. REM sleep is characterized by wake-like patterns in the EEG, often (but not always) relaxed skeletal musculature from the preceding non-REM sleep level, eye movements behind closed eyelids and behavioral restfulness. As illustrated by the example postures of pigeons and Australian magpies, behavior can give some insights into a bird's state, but it can be difficult to distinguish between sleep states or even between sleep and wakefulness [42]. These EEG, EMG and accelerometry traces are representative examples recorded from an Australian magpie. Illustrations by Laura X. Tan.

## 3. Light Regulates and Suppresses Sleep in Diurnal Birds

Like most organisms, birds have evolved to keep time with a predictable 24-h cycle of light and darkness. Therefore, light has a fundamental influence on behavior and physiology. The effect of light on sleep can be both indirect and direct (Figure 2). Light can indirectly affect sleep by shifting the time of the circadian clock. Light is detected by photoreceptors in the retinas (melanopsin photoreceptors) and also penetrates through the skull to photoreceptors in the brain, including in the pineal gland (pinopsin) and hypothalamus (vertebrate ancient opsin) [45,46]. Melanopsin, pinopsin and vertebrate ancient opsin are most sensitive to 480 nm, 460 to 470 nm, and 460 nm of light, respectively [47]. Detection of light by these photoreceptors can alter the expression of clock genes and suppress melatonin, a hormone that is important for synchronizing (or entraining) the internal circadian clock with the external light–dark cycle [45,47–49]. These genetic and hormonal pathways are thought to mediate effects of light on avian sleep [45,50]. In addition to these indirect effects, light at night can directly suppress sleep (without necessarily shifting the circadian clock) by masking natural light–dark cycles. One way this may occur is by allowing diurnal animals to see more at night, presenting different opportunities and threats. These include increased opportunities to forage [51] and mate [52], as well as greater risk of predation. Introducing light to unusual times and places might also act

as a novel visual stimulus that increases alertness. Currently, we know very little about the relative importance of these various pathways or how these effects interact. Still, it is helpful to consider these possible mechanisms when contemplating effects of light on sleep.

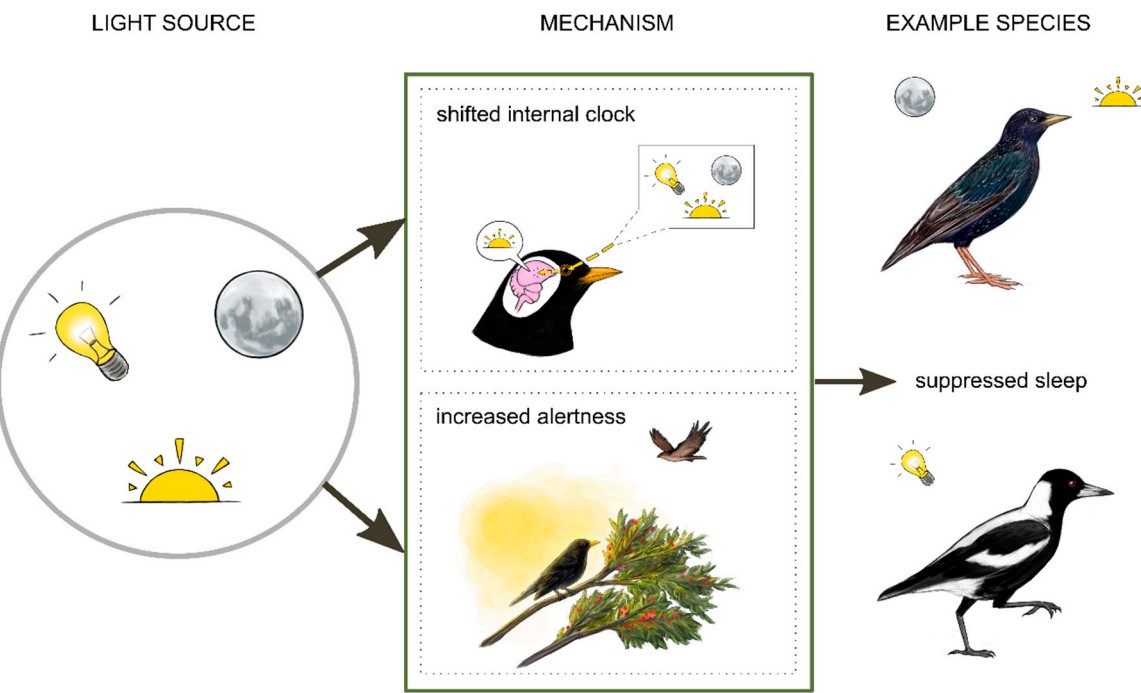

**Figure 2.** Exposure to light can suppress sleep in birds. European starlings sleep less in summer and during a full moon [53]. Australian magpies show disrupted sleep under artificial light at night [54]. There are two broad mechanisms by which these effects might occur. Light might influence sleep indirectly by shifting the timing of the circadian clock. In European blackbirds, increased activity at night during light exposure has been linked to reduced melatonin, a hormone important for regulating circadian timing [55]. Light might also directly increase alertness by masking natural light cues and facilitating foraging or vigilance [51]. Illustrations by Laura X. Tan.

The first insights into how light regulates avian sleep came from laboratory studies of pigeons (*Columba livia*). Early research suggested that continuous light exposure virtually eliminated sleep in pigeons (n = 2) for weeks with no subsequent rebound in the amount or intensity of sleep [56]. This interpretation is disputed [39,57,58]. Numerous studies have since demonstrated that while pigeons sleep mostly at night, they nonetheless nap throughout the day. Moreover, daytime sleep is homeostatically regulated, casting doubt on the conclusions reached by those early studies [39,57,58]. Another study found that when moved from 12:12 to 3:3 light:dark (LD) cycles, pigeons still slept primarily during the dark phase (based on eye closure) [59]. This suggests that light has an alerting effect independent of the internal clock. However, pigeons slept less overall when exposed to 3:3 LD cycles, indicating that circadian timing influences sleep.

Natural variation in daylength can also influence sleep in birds. Recent studies have found that European starlings (*Sturnus vulgaris*) and barnacle geese (*Branta leucopsis*) sleep less in summer than in winter [53,60]. Starlings had 5 h less non-REM sleep (per 24-h day) in summer, as well as a reduced proportion of REM sleep out of total sleep [53]. Barnacle geese had 1.5 h less non-REM sleep in summer but showed no difference in REM sleep [60]. These reductions in sleep, although substantial, were not commensurate to changes in daylength, which ranged from around 9 to 20 h. During summer, both starlings and geese also had more fragmented sleep and slept more during the day. These findings are generally consistent with behavioral studies of blue tits (*Cyanistes caeruleus*). Steinmeyer et al. [61] and Schlicht and Kempenaers [62] found that blue tits slept 4–5 h less in spring than in winter, based on sleep behavior and time spent in the nest box, respectively. The timing

of sleep also varied seasonally; in autumn and spring, blue tits usually began sleeping before sunset, whereas in winter, they began sleeping after sunset. Intriguingly, there is also evidence for seasonal variation in the homeostatic regulation of sleep. When barnacle geese were deprived of sleep for 4 or 8 h, they showed a compensatory rebound in non-REM sleep amount during summer, but not during winter [60]. These findings may not be due to photoperiod alone; other variables, such as temperature and vulnerability to predation, might also play a role. Nevertheless, these findings suggest that photoperiod might not only influence the amount, timing and composition of sleep in birds, but also the extent to which birds can tolerate sleep loss.

Recent research indicates that moonlight can suppress sleep in birds. According to van Hasselt et al. [53,60], European starlings and barnacle geese had 2 h less non-REM sleep during a full moon compared with a a new moon. The findings for geese were partially supported by a subsequent study [63], which found decreased non-REM sleep during a full moon in winter, but not in summer. In another study conducted during summer, Aulsebrook et al. [44] also found no effect of the lunar phase on the total amount of sleep in black swans (*Cygnus atratus*). This latter study on swans used activity as a proxy for sleep (non-REM and REM sleep combined) after validation with recordings of the EEG, such that it was not possible to distinguish between non-REM and REM sleep. There is some evidence that moonlight can increase activity in some birds; for example, diurnal willie wagtails (*Rhipidura leucophrys*) sing more at night during a full moon [64]. Greater exposure to evening light (from unspecified sources) has also, counterintuitively, been associated with increased and more continuous sleep behavior in great tits [65]. However, further research would be required to determine whether moonlight affects sleep in these two species.

Given that photoperiod and moonlight can influence sleep, it is perhaps unsurprising that artificial light can as well. When exposed to urban intensities of white light at night, sleep in captive pigeons and Australian magpies (*Cracticus tibicen*) was reduced, less intense (reduced slow-wave activity during non-REM sleep), more fragmented and had a lower fraction of REM sleep (out of total sleep) relative to a dark night [54]. Moreover, not all of the sleep that was lost was subsequently recovered. Increased exposure to artificial light at night has also been associated with reduced sleep in captive barnacle geese [63]. In this study, exposure to light was mediated by cloud cover, with artificial light reflecting back off the clouds. On moonless nights, geese had less non-REM sleep when there was medium-to-maximum cloud cover. However, during a full moon, sleep in geese did not vary with cloud cover, likely because the light from artificially illuminated clouds was similar to moonlight. Numerous studies also found that when exposed to artificial light at night, birds show increased activity [44,55,66–73], delayed chirping behavior [74], increased vigilance [75] and reduced sleep behavior at night [43,76–79], but see [80,81]. There is some evidence that these changes in behavior are associated with reduced circulating oxalate [66,79], a biomarker of sleep debt in rats and humans [82]. However, other studies found an increase [83] or no change [84] in oxalate under artificial light at night, raising the possibility that the level of oxalate does not reflect prior sleep/wake history in birds [84]. In captive zebra finches (*Taeniopygia guttata*), artificial light at night has also been found to alter expression of genes involved in sleep regulation [79]. Together, these studies provide accumulating evidence for disruptive effects of artificial lighting on avian sleep.

For some, but not all, bird species, the effects of light on sleep seem to depend on the spectral properties of the light. Lights that emit more blue light are broadly predicted to be worse for sleep, largely because non-visual photoreceptors are most sensitive to blue light (short-wavelength light in the blue region of the visual spectrum) [22,85,86]. Accordingly, in Australian magpies, blue-rich light at night was more disruptive for sleep than blue-reduced light [54]. However, pigeons in the same study showed no difference between the two types of light. Likewise, in captive black swans, blue-rich and blue-reduced lighting caused similar reductions in nighttime sleep [44]. Behavioral studies have produced similarly mixed findings. Captive zebra finches were more active at night under

blue-rich light compared with blue-reduced light [70]. However, a study of captive blue tits found that red (blue-reduced) and white light caused similar increases in nighttime activity compared with green (blue-rich) light or darkness [73]. Wild great tits roosting near red (blue-reduced) and green (blue-rich) lights also showed similar activity levels at night, which were lower than those of tits roosting near white light [66]. In wild great tits, the effects of light spectra also seemed dependent on the birds' origin; forest birds were more active under white light than green light, whereas urban birds showed no difference [84]. Consequently, adjusting light spectra may be more beneficial for sleep in some species and contexts than in others.

Why is there such variation in the responses of birds to light? The answer to this question remains unclear, although there are a few possible interpretations. One possibility is that these differences are driven by confounding disparities in lighting between studies. For example, more intense lighting might have greater effects on avian sleep irrespective of light spectra. However, different light intensities cannot explain the contrasting effects of blue-rich and blue-reduced lighting on Australian magpies and pigeons (see [54] for an in-depth discussion), suggesting that there are other factors at play. A second possibility is that species vary in their behavioral responses to light due to their visual capacity or ecology. For example, some birds might be at greater risk of predation when exposed to light at night, while others are not. There is evidence that some birds choose to roost near lights [84], while others avoid them [87]. Such behavioral differences might also influence the physiological effects of light on sleep if some birds are more exposed to light at night than others. Finally, there may be physiological variation in the responses of birds to light, perhaps driven by differences in evolutionary history. Some birds, including migratory birds, may be better adapted to changes in ambient light regimens than others and therefore respond differently. To better understand the variation in effects of light on avian sleep, we therefore need a better understanding of the underlying mechanisms.

Existing studies of how artificial light affects avian sleep can provide some clues about underlying mechanisms. For example, Spoelstra et al. [88] found that dim light at night (0.15 to 5 lux) directly shifted the timing of activity in great tits without shifting the timing of the circadian clock. This suggests that the change in activity was driven by a direct alerting effect of light on behavior. The effects on sleep immediately after nighttime lighting is switched off can also offer insights. Australian magpies rapidly recovered non-REM sleep after 4 h of exposure to white or amber light, implying that these effects were acute and direct [54]. However, in pigeons, non-REM sleep intensity continued to be reduced more than 24 h after exposure to white light throughout the 12-h night. The peak in non-REM sleep intensity also appeared delayed, which further suggests some physiological disruption of sleep regulation. Other research demonstrates that urban intensities of light at night (0.03 to 8 lux) can suppress circulating melatonin concentrations [55,68,72,89] and alter expression of clock genes [79], suggesting that circadian pathways could mediate some of the effects of artificial light at night on sleep. Together, these findings suggest that artificial light can influence sleep both directly and indirectly, and that the importance of various pathways might vary depending on the species or context.

Overall, growing evidence demonstrates that daylength, moonlight and artificial light at night can influence the total amount, timing and structure of sleep in birds. However, most of our understanding comes from a handful of EEG-based studies of captive pigeons, European starlings and Australian magpies. Behavioral studies also offer valuable insights and do not require an invasive surgical procedure, making data easier to collect with less imposition on the animals. Still, behavioral studies can miss important effects on sleep composition, intensity and subsequent recovery [41,54]. Future research should also be careful to consider the actual illumination of a bird's environment, which might be mediated by factors such as roosting site and cloud cover [63,81]. Nevertheless, the big question that remains is: do these effects of light on sleep matter for avian performance?

## 4. Sleep Affects Avian Performance

Sleep can benefit almost all aspects of animal biology, including the maintenance of the brain and central nervous system [90,91]. Consequently, sleep is generally thought to be important for waking performance. Research on birds indicates that, as in mammals [32], sleep is beneficial for optimal cognitive functioning (Figure 3). The bulk of research in this area has been of learning processes during development, such as imprinting and song learning.

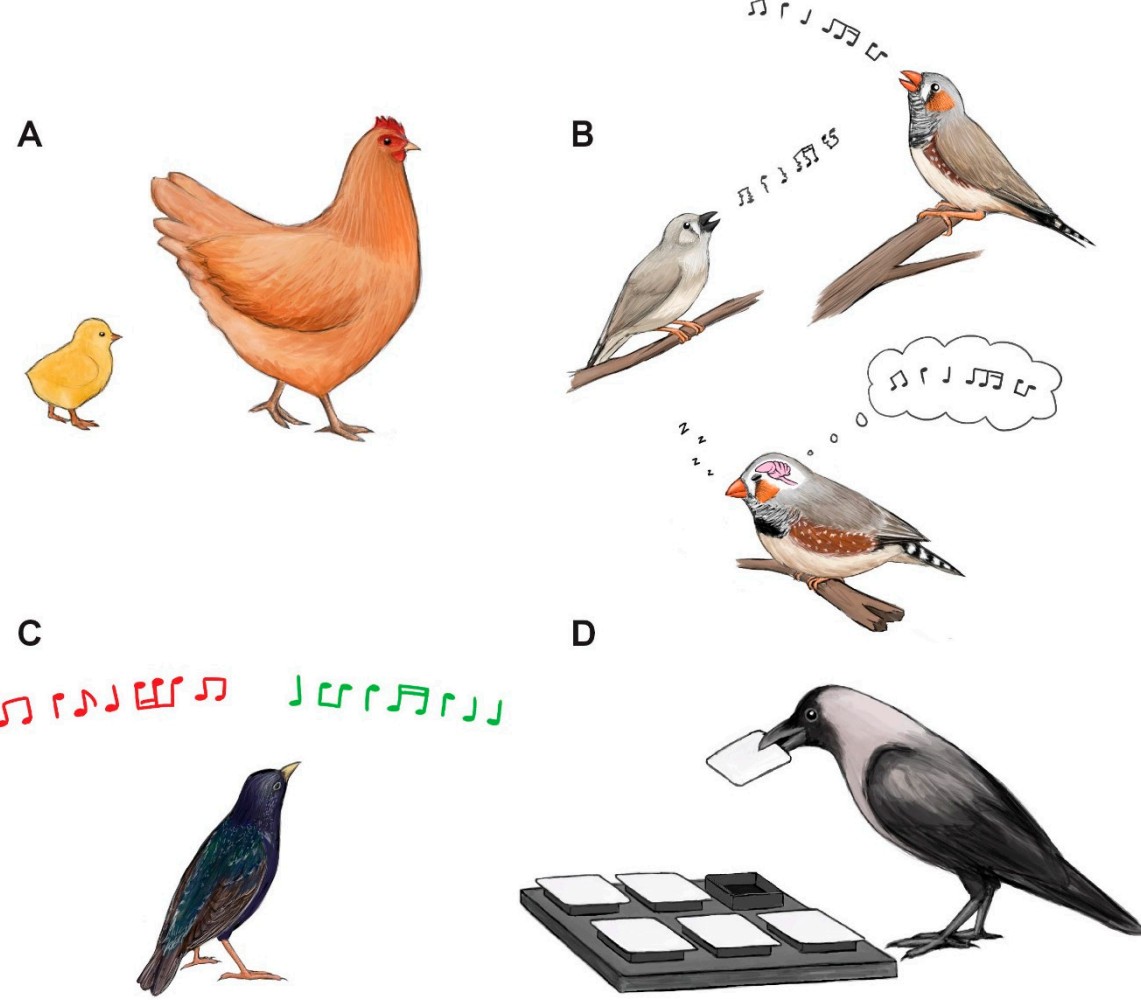

**Figure 3.** Multiple cognitive processes in birds depend on sleep, including: (**A**) imprinting in domestic chicks, (**B**) developmental song learning and neuronal replay in zebra finches, (**C**) auditory discrimination learning in European starlings and (**D**) spatial learning in Indian house crows. Illustrations by Laura X. Tan.

### 4.1. Imprinting

Filial imprinting is a social recognition and bonding response, where a young animal is exposed to an object, then selectively follows that same object. In the wild, the object of attachment would be a conspecific, usually the mother [92,93]. In young birds, imprinting behavior is important for survival, since the parent is often the only source of food and protection. Memory for imprinting is quickly formed and encoded in the left intermediate and medial mesopallium (IMM), then redistributed to a (so far unidentified) network called S' for long-term storage [92,94].

Studies of chicks (*Gallus gallus domesticus*) indicate that sleep is important both for encoding and consolidating imprinting memories. In the IMM, the largest increase in

neurons selective to the imprinting stimulus occurs after chicks sleep [95]. Sleep also maximizes the proportion of neurons responsive to the imprinting stimulus and stabilizes the selective responses of these neurons. For this process to be effective, sleep must occur shortly after training on the imprinting stimulus [96]. Furthermore, the distribution of imprinting memories to long-term storage seems to take place during sleep and takes many hours to manifest [94,97–99]. This pattern of immediately strengthening a memory, then redistributing and encoding it for long-term storage is remarkably similar to processes observed during non-REM sleep in mammals [100]. In one study, chicks showed an increase in 5–6 Hz activity when sleeping after imprinting training [96]; however, the significance of this increase remains unclear. REM sleep might also contribute to learning, with one study finding that chicks spent more time in REM sleep and had more REM sleep episodes after imprinting [101].

*4.2. Song Learning*

In passerines (songbirds), sleep can be important for learning elaborate vocalizations called songs. Songbirds use their songs to communicate, including to find mates and defend territories. They develop their songs at an early age by first forming a template of a tutored song, then practicing by imitation and auditory feedback until the song becomes an accurate imitation of the tutor song. By the time of sexual maturity, the song crystallizes, which is characterized by stereotypy [102–104].

During development, sleep influences song structure. Across the day, as birds practice singing, their songs become more complex and similar to the tutor song. However, song complexity declines across sleep [13]. This decline in complexity occurs even when sleep is induced during the day, revealing a role for sleep per se rather than circadian modulation of song quality. Interestingly, juvenile zebra finches that mimic the tutor song most accurately in the long term are also those that show the greatest deterioration in song complexity after sleep. This might seem counterintuitive, considering how sleep generally enhances learning and memory consolidation in mammals. However, these phenomena could be related to development and age. Early in development, juvenile vocalizations are highly variable, and song learning involves extensive daily practice. Thus, reduced song complexity after sleep could help prevent juveniles from crystalizing inaccurate song features too early in life [103].

Sleep also influences song learning at a neuronal level. There are several brain regions that make up the avian song system, which have been studied extensively in zebra finches. During sleep, neurons in these regions spontaneously activate in a pattern that matches activity during daytime singing [105]. This phenomenon is known as neuronal replay and occurs in premotor areas, such as the HVC and the robust nucleus of the arcopallium (RA) [105–107]. During sleep, juvenile birds with the greatest song deterioration exhibit the lowest firing rate in the HVC. In addition, HVC firing rates increase across development, which might help to stabilize the maturing song [108]. Bursting spike activity in RA neurons also increases during sleep after the birds have listened to the tutor song during the day, but only if the birds listened to their own singing afterwards, suggesting that auditory feedback is an important mechanism for song learning [109]. In addition, spontaneous neuronal activation during sleep has been found in the brain area responsible for auditory association, the caudomedial nidopallium (NCM), although only in juvenile birds. This neuronal activity during sleep is linked to the amount of tutor song stimulation and accuracy of song performance during prior wakefulness [110].

*4.3. Auditory Discrimination*

Communication relies on the interpretation of information by a receiver. In a natural setting, birds have to learn to discriminate between a wide array of different songs and determine which ones are relevant. Furthermore, exposure to new songs can interfere with existing memories of similar songs. Such interference can impair a bird's ability to recall and discriminate songs accurately [111].

Sleep can not only help birds discriminate between novel songs, but also restore memories that have been impaired by interference. Brawn et al. [112] found that European starlings were better at discriminating between novel conspecific song segments after both nocturnal and daytime sleep. In a second study, the researchers trained birds to recognize an interfering song segment similar to a previous song segment [111]. Birds that were kept awake between training and testing had impaired memories of the original and interfering song segments compared with birds that were given the opportunity to sleep. Starlings that encountered interference before sleep showed significant improvements after sleep, demonstrating that sleep restored and enhanced classification memory. In addition, interference after sleep had no effect on the previous day's learning, suggesting that sleep has a protective effect against the effects of interference on memory.

A more recent study suggests that sleep is also important for reconsolidation of retrieved song memories. When auditory memories are retrieved, they return to an unstable state, making them vulnerable to interference. However, Brawn et al. [113] found that during sleep, the destabilized memories were reconsolidated through a cycle of use-dependent destabilization and sleep-dependent reconsolidation. This cycle repeated across several days, while overall song discrimination performance improved. More importantly, however, when birds encountered an interference song on the day after learning the initial song, their performance of the initial song improved more after sleep than when the interference song was not encountered. Therefore, the interference song seems to have enhanced the memory for the initial song through a sleep-dependent mechanism. The reason for this effect could be that learning the interference song reactivated the memory of the initial song, i.e., if the songs activated similar networks within the brain [113]. In the wild, such as when a juvenile songbird is learning a specific song, reconsolidation might help stabilize that memory in the presence of interfering songs from other birds [114]. Reconsolidation during sleep may also help to refine the complexity of vocalizations learnt during the sensorimotor phase, when a juvenile is using auditory feedback to refine its song, which would result in a more accurate copy of the tutor song [13,109].

### 4.4. Spatial Learning

There is some preliminary evidence that sleep affects spatial learning in birds. For wild birds, remembering different locations such as home range and (in the case of food-storing birds) the location of food caches is extremely important for survival, especially in harsh environments [115–117]. In a study of chickens, Nelini et al. [118] found that chicks spent more time asleep after learning a spatial discrimination task. Further research indicated a link between closing one eye during sleep and spatial learning [118,119]. The authors suggested that this could mean that the benefits of sleep for spatial learning are localized to specific parts of the brain. There is also evidence that increased activity during exposure to constant light, is associated with poorer spatial and pattern learning in adult Indian house crows (*Corvus splendens*; see "Evidence linking light, sleep and performance in birds") [120].

Taken together, the studies discussed above reflect the importance of sleep in maintaining performance in awake birds, reflected at neuronal and behavioral levels. To better understand these processes, future research should focus on whether different sleep stages and specific spectral features of brain activity contribute to different functions. In addition, these studies have been conducted exclusively in a laboratory context, often with more severe deprivations of sleep than might occur in a natural setting. How sleep influences performance in real-world contexts is largely unclear. Could sleep loss incurred in the wild, such as through exposure to light pollution, affect avian learning and memory? Does impairment in cognitive performance matter for survival and reproduction?

## 5. Evidence Linking Light, Sleep and Performance in Birds

Studies linking ambient light to sleep and performance are uncommon. Nonetheless, those few studies have (1) revealed previously unrecognized intraspecific variation in sleep amount, (2) demonstrated that some animals are resilient to the oft-reported negative

impacts of sleep loss on waking performance and (3) challenged prevailing views on the adaptive value of sleep. Below, we highlight, in our opinion, the most salient studies. We begin with studies that either did not directly measure sleep or used a behavioral characterization of sleep, as EEG-based studies are few.

Artificial light at night appears to have little effect on problem-solving in peafowl (*Pavo cristatus*). Adult birds were exposed to artificial light at night for a single night, then presented with a bowl of mealworms to eat the following morning [121]. An adjacent bowl also contained mealworms, but those were obscured by translucent paper. The birds were required to peck through the paper to access the mealworms beneath. It was thought that birds would solve the test poorly after a night under lights, owing to increased nighttime vigilance (based on eye closure) [75]. However, performance was unaffected. Nonetheless, it remains to be demonstrated that the peafowl had a disrupted night of sleep prior to the test, and if they had, whether they simply recovered lost sleep by sleeping more intensely. Alternatively, the task may have been simple enough for the birds to solve even after sleep deprivation.

This limitation was addressed, in part, by a sleep study of great tits. Here, Ulgezen et al. [84] characterized sleep behaviorally, based on posture (head tuck) and restfulness (no movement in the preceding minute). The birds were caught in the wild and exposed to artificial light at night for 11 nights in captivity. During the light treatment, birds were also tested on learning and memory tasks. Interestingly, artificial light at night disrupted sleep, yet the birds preferred to sleep under lights rather than in darkness. In addition, despite the disruptive influence of night lights on sleep, task performance was unimpaired. Thus, as with the peafowl, light at night appears to have little influence on subsequent waking performance. Nonetheless, given the behavioral nature of the study, it remains to be investigated whether birds could recover lost sleep by sleeping more intensely. As noted by the authors, sleep may simply have not been sufficiently disrupted to impair performance. This idea has some support owing to the lack of predicted response in oxalic acid, a biomarker of sleep loss in mammals [82].

In contrast to peafowl and great tits, Indian house crows showed impaired performance on spatial and pattern-association learning tasks when exposed to constant light [120]. Interestingly, constant light exposure reduced the activity in dopamine neurons in the midbrain, which have been implicated in learning and memory in mammals [122,123]. This reduction in dopamine coincided with a decline in neuronal activity and neurogenesis in the hippocampus and caudal nidopallium [120]—brain regions associated with learning and advanced cognition in birds, with the latter region thought to be the avian equivalent of the prefrontal cortex. Reduced performance could have been due to sleep loss, as crows became more arrhythmic [120] and have been found to have reduced restfulness under constant light [124]. Consequently, although low levels of light at night might not influence performance, these studies suggest that constant light or extended photoperiods might negatively affect performance by disrupting sleep.

In natural contexts, some birds show seasonal variation in sleep, which can offer additional insights. Most songbirds are day-active outside of migration, but switch to migrating at night during the breeding season. Even in captivity, the drive to migrate manifests as migratory restlessness (or Zugunruhe) with the birds flapping their wings and hopping towards the direction of their intended migration. The white-crowned sparrow (*Zonotrichia leucophrys gambelii*) migrates twice per year between Alaska and southern California. In a long-term laboratory-based study, white-crowned sparrows were studied over the course of a year [125]. As expected, the sparrows increased their nighttime activity in the spring and autumn, just like they would in the wild. Recordings of the EEG confirmed that birds slept 63% less at night (and as much as 85%) when in a migratory state. Although birds became drowsier during the day, they did not recover non-REM sleep by sleeping more intensely at night, nor did they show consolidated periods of unequivocal non-REM sleep during the day.

There is some evidence that seasonal variations in sleep predict cognitive performance. Cognitive performance was assessed by training each sparrow on an operant task, which required the birds to peck keys in sequence to obtain a seed reward (for details see [125]). When non-migratory birds were deprived of sleep in a manner emulating sleep restriction in migratory birds, their performance on the task declined. Conversely, the performance of sparrows in a migratory state remained high. It is unclear how these birds maintained cognitive function with so little sleep. Nonetheless, as noted by the authors, stress associated with the induced restriction of sleep in non-migratory sparrows may have contributed to their reduced performance on the task. Furthermore, different tasks may yield different outcomes. Migratory sparrows are more impulsive than non-migratory sparrows; this impairs their performance on tasks where elevated response rates reduce access to food [126]. Studies that assess performance in the wild would remove confounds associated with stress and laboratory housing. It is worth mentioning that the severe reduction in sleep occurred without lengthening the light phase of photoperiod, indicating that sleep restriction arose from endogenous changes in migratory state. Still, these results imply that seasonal changes in photoperiod in the wild might induce reductions in sleep without negatively affecting performance.

Another bird that shows reduced sleep under extended photoperiods without reduced performance is the pectoral sandpiper (*Calidris melanotos*). Pectoral sandpipers are small shorebirds that overwinter in the Southern Hemisphere. During spring in the Northern Hemisphere, sandpipers migrate to above the Arctic circle to breed. The breeding season is short and extremely competitive. Females scrutinize males to select the best mate to sire her only clutch of the season. Conversely, males have just three weeks to sire as many offspring as possible. Under continuous daylight of the midnight sun, males with increased activity (and reduced sleep) would have more time to court females, deter rivals, watch for predators and forage. Indeed, a multi-year study found that some male sandpipers were extremely active; in the most extreme case, one male was active more than 95% of the time for 19 days [6]. Such extremes were specific to males, and males were more active than females throughout the breeding season. Only once fertile females were no longer present, owing to females incubating eggs, did males decrease their activity.

Further investigation revealed that more active male pectoral sandpipers were not only sleeping less, but also performing better [6]. To confirm that increased activity reflected reduced sleep, the EEG was recorded from a subset of males on the tundra. These recordings showed that some, but not all, males slept little and for short periods of time, best measured in tens of seconds. Nevertheless, these males accrued a sleep debt, as revealed by more intense non-REM sleep during those brief naps. Despite restricted and fragmented sleep, shorter-sleeping males ultimately interacted with more females and sired more offspring (Figure 4). However, not all males became superactive during the breeding season. The constant light condition of the high Arctic summer therefore does not cause males to become superactive, but rather allows some males to perform well round-the-clock. Why only some males were capable of this extreme behavior remains unclear. Nevertheless, in a real-world environment and using the most important metric of performance from an evolutionary perspective, i.e., reproductive success, sleep loss was shown to be evolutionarily adaptive.

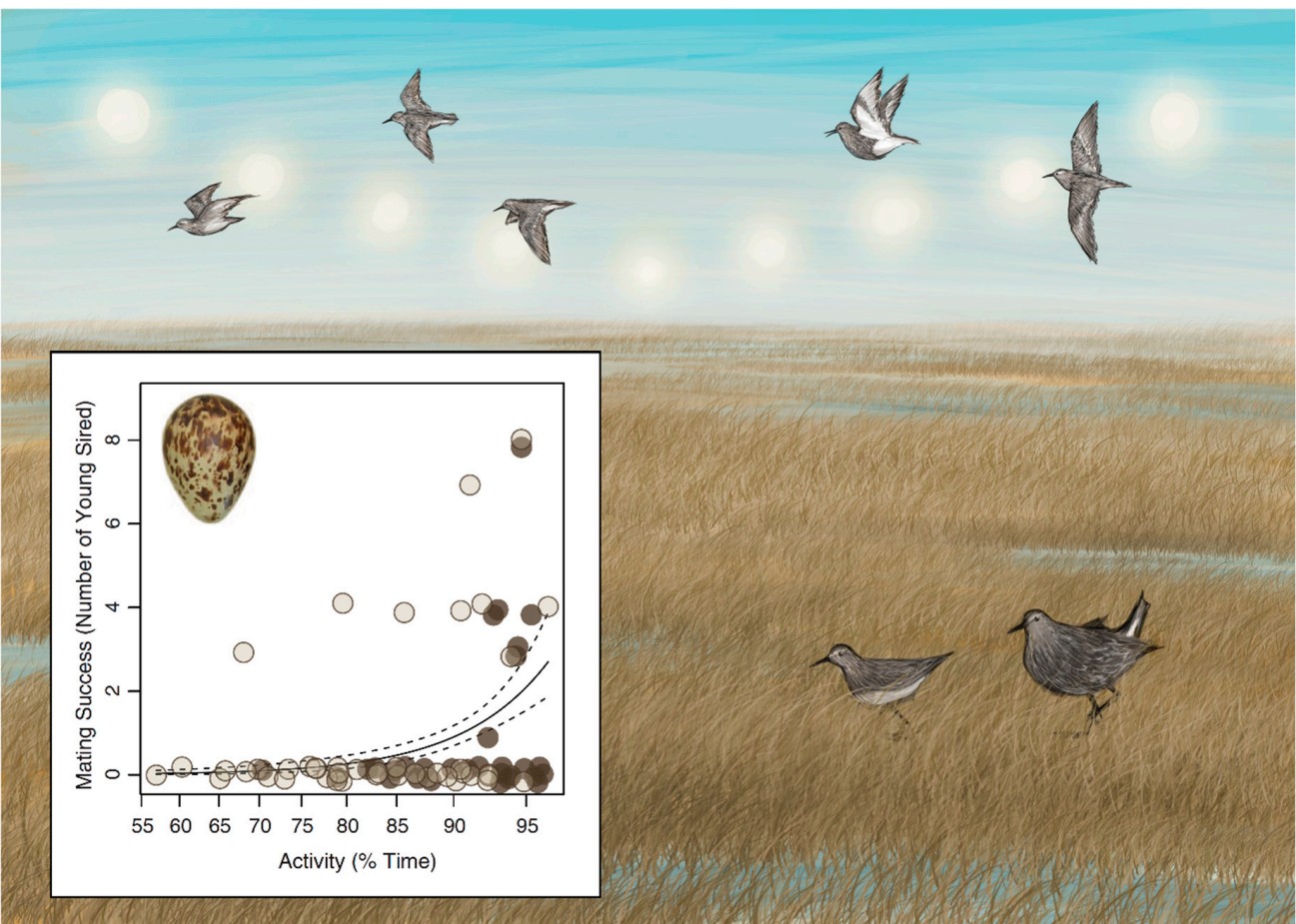

**Figure 4.** The sun never sets during summer in the high Arctic. In this environment, some male pectoral sandpipers sleep little and are extremely active during the three-week breeding season. Although there was substantial male–male variation in the level of activity, males that slept the least ultimately sired the most offspring (inset from Lesku et al. [6]). Illustrations by Laura X. Tan.

## 6. Conclusions and Future Directions

In birds, there is accumulating evidence that light affects not only the amount of sleep, but also the structure, timing and regulation of sleep. There is also evidence that sleep influences waking performance, especially learning and memory during development. However, there are still crucial gaps in our understanding. Although artificial light at night can disrupt sleep, we do not know if these disruptions are sufficient to impair performance. While sleep can influence cognitive performance, it is also uncertain whether this has implications for survival and reproduction. Furthermore, only a handful of studies have directly linked light exposure to both sleep and performance. Most of these do not support the intuitive narrative. Instead, multiple studies find that birds can perform just as well or even better with restricted sleep. These findings continue to baffle our understanding of the adaptive value of sleep for these birds.

The relationship between light, sleep and performance in birds presents a valuable opportunity for research. The topic is important for our understanding of how sleep evolved in environments that are characterized by changes in light with time of day and season. Even in the past year, we gained new insights into the incredible variation in avian sleep in response to natural light cycles. Given the unprecedented pervasiveness of artificial light at night worldwide, this research topic is also important from a conservation perspective. Untangling the various mechanisms and consequences of varying light exposures for sleep

and subsequent waking performance could prove instrumental for our understanding of animal behavior.

**Author Contributions:** Conceptualization, A.E.A., R.D.J. and J.A.L.; investigation, A.E.A., R.D.J. and J.A.L.; writing—original draft preparation, A.E.A., R.D.J. and J.A.L.; writing—review and editing, A.E.A., R.D.J. and J.A.L.; visualization, A.E.A., R.D.J. and J.A.L. All authors have read and agreed to the published version of the manuscript.

**Funding:** This research received no external funding.

**Institutional Review Board Statement:** Not applicable.

**Informed Consent Statement:** Not applicable.

**Acknowledgments:** The authors thank Laura X. Tan for providing illustrations for the figures in this review. The authors also thank Farley Connelly, Michelle L. Hall, Therésa M. Jones and Raoul A. Mulder for their valuable comments on earlier versions of part of the manuscript.

**Conflicts of Interest:** The authors declare no conflict of interest.

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
