# Peer review of "Light, Sleep and Performance in Diurnal Birds"

_2624-5175, doi:10.3390/clockssleep3010008_

Round 1

Reviewer 1 Report

This is a nice review and was very educational for a non-avian researcher such as myself. The writing style was particularly easy to read, which I appreciate. I hope the below comments are useful in refining the manuscript.

Line 57: As this is a review, can you provide example values so that the reader doesn’t have to dig them up themselves? It would also be very interesting to know if avian sleep exhibits other features of sleep as seen in humans such as k complexes, sleep spindles, and decreasing delta sleep over time.

Line 67: “presently too small” - I think the phrasing here needs adjusting (unless you expect these birds to change in size!).

It would be really helpful to include a figure that shows behavioral movement over time (multiple days) in some bird species. I am used to seeing obvious and pronounced quiescence/sleep consolidation in human actigraphy data and rodent wheel-running data. I assume this varies considerably by bird species and some depictions of this variability would enhance the manuscript. It would also be helpful in understanding the phase angle between sunlight hours and bird sleep and how it differs by bird.

Line 78: You should probably explicitly state that light is able to penetrate the skull to be sensed by these in-brain photoreceptors. There’s a lot of research looking at modifying the wavelengths of light to less impact circadian rhythms in humans, these intrinsically photosensitive retinal ganglion cells (ipRGCs) are maximally sensitive to ~480 nm light that is prevalent in natural sunlight. It would be useful to know what the spectral sensitivity of these bird photoreceptors are.

Line 80: (Unless things are different in birds), melatonin does not entrain the circadian clock in the first instance. Rather, its expression is dependent upon the phase of the circadian oscillator and melatonin in-turn regulates peripheral oscillators (we think). Also, is melatonin expressed at night in nocturnal birds still, or is it expressed during light in these species?

Line 107: From the writing, it’s not clear who Berger and Phillips are, is this citation 46?

Line 122: Can you state what the daylength (sunlight duration) was for each species? I.e., is the reduction in sleep commensurate with the increased daylength in summer?

Line 137: What about REM sleep duration, was it affected?

Line 140: Do they sing more at night or on the subsequent day (I thought wagtails were diurnal but I could well be wrong…)?

Line 147: Aren’t less intense and lower fraction of REM sleep the same thing? You might want to clarify what you mean by intensity, or, better yet, remove the term entirely.

Line 159: An important confound not addressed here is the intensity of the artificial lights, which I guess vary across the studies cited. It is generally agreed in terms of the circadian system that shifting intensity can have a bigger effect than shifting spectrum. This is an important point to make the reader aware of.

Line 172: This is very surprising to me; can you state what the likely reason for this is?

Line 176: So, are we to infer that the effect is due to increasing alertness, rather than shifting the clock?

Line 179: But not REM sleep?

Generally, it’s very surprising to me that the effects of light are so variable by avian species. Is there some hypothesis as to why this is? E.g., is it due to the temporal niche they occupy (diurnal vs nocturnal)? Is there a bodyweight/brain size relationship? Or are we to expect the physiology differs that greatly between species (I would’ve thought that unlikely). Given the nature of this review, a discussion of the source(s) of such variability would be useful.

Line 376: It’s hard to interpret what constitutes a “great reduction in sleep” so a bit more detail would be good.

Figure 4: I’m sad to say this wasn’t visible on my review copy of the manuscript.

Reviewer 2 Report

The paper is interesting since relation between light, sleep and performance is an actual argument of debate. In the review relevant studies are present, adequately categorized and summarized. The paper is well organized and written making reading pleasant. The review can be helpful to future studies that will investigate the mechanism linking light and sleep not only in birds but also in mammals.

The manuscript related the influence of light on the internal circadian clock citing the effect on melatonin levels. However, several studies have shown that not all light radiations are capable of having an effect on the internal biological clock and, moreover, the relation between clock genes and circadian clock is being studied for several years. One or more paragraphs focusing on the blue light and clock genes could be useful to enrich, update and complete the paper.

The format of lines 74-87 need revision.

Reviewer 3 Report

I studied this manuscript with interest and found much useful information, it is comprehensive enough and well-written. I believe it can be accepted in the present state.

Few comments, the manuscript currently is entitled “Light, sleep, and performance in birds”. Authors refrained from a comprehensive review of light of different phase structures and duration of the natural diurnal phase, in particular from discussing photoperiodic seasonal and latitudinal photoperiodic factors, they are discussed only briefly.

Also, on L.46 authors wrote: “As little is known about these (nocturnal light) effects on nocturnal birds, we necessarily focus on birds that sleep predominantly at night”. Thus, the features of the sleep of nocturnal bird species were left off-scope. In fact, the authors mainly discuss the effects of artificial light at night in diurnal birds, therefore changing the article title to "Light at night, sleep and performance in diurnal birds” might be justified.

Discussion of responses of light on sleep in migratory and non-migratory species combined can be too complex and even sometimes misleading since migratory species apparently have a solid evolutionary background of adaption to cycling changes in ambient light regimens that deserves in-depth analysis.

Please note that Figure 4 is not depicted.

Round 2

Reviewer 2 Report

The additions that have been made in the paper are small, and not all requests have been satisfied. However, the quality of the paper is enough.